# Dusky-like Is Critical for Morphogenesis of the Cellular Protuberances and Formation of the Cuticle in *Henosepilachna vigintioctopunctata*

**DOI:** 10.3390/biology12060866

**Published:** 2023-06-15

**Authors:** Yuxing Zhang, Qiao Tan, Mengjiao Lin, Chenhui Shen, Lin Jin, Guoqing Li

**Affiliations:** Education Ministry Key Laboratory of Integrated Management of Crop Diseases and Pests, State & Local Joint Engineering Research Center of Green Pesticide Invention and Application, Department of Entomology, College of Plant Protection, Nanjing Agricultural University, Nanjing 210095, China; 2020202042@stu.njau.edu.cn (Y.Z.); 2020102083@stu.njau.edu.cn (Q.T.); 2020102082@stu.njau.edu.cn (M.L.); 2019202031@njau.edu.cn (C.S.); jinlin@njau.edu.cn (L.J.)

**Keywords:** *Henosepilachna vigintioctopunctata*, dusky-like, bristle, scolus, development

## Abstract

**Simple Summary:**

In *Drosophila melanogaster* and *Tribolium castaneum*, dusky-like (Dyl) is related to linking the insect cuticle to the plasma membrane of epidermal cells, regulating the cytoskeleton and deposition of the cuticle. A deficiency in Dyl impairs the formation of embryonic denticles and adult sensory bristles and wing hairs. In a polyphagous beetle, *Henosepilachna vigintioctopunctata*, we characterized *Hvdyl* using RNAi. The results showed that the knockdown of *Hvdyl* inhibited the growth of cellular protuberances, inhibited foliage consumption, and affected the survival of the beetles. Our findings indicated that two functions of Dyl in *Drosophila* are conserved in *H. vigintioctopunctata*.

**Abstract:**

Dusky-like (Dyl) is a transmembrane protein containing a zona pellucida domain. Its physiological roles during metamorphosis have been well explored in *Drosophila melanogaster* and have also been documented in *Tribolium castaneum*. However, Dyl has undergone a functional shift between Diptera and Coleoptera insects. Further investigation of Dyl in other insects will be helpful to further clarify its function in insect growth and development. *Henosepilachna vigintioctopunctata* is an important Coleoptera that causes enormous economic losses in agriculture in China. In this study, we found that the expression of *Hvdyl* was detectable in embryos, larvae, prepupae, pupae, and adults. We knocked down *Hvdyl* in third- and fourth-instar larvae and pupae with RNA interference (RNAi). RNAi of *Hvdyl* mainly caused two phenotypic defects. Firstly, the growth of epidermal cellular protuberances was suppressed. Injection of ds*dyl* (double-stranded *dusky-like* RNA) at the third-instar larval stage truncated the scoli throughout the thorax and abdomen and shortened the setae on the head capsules and mouthparts of the fourth-instar larvae. Introduction of ds*dyl* at the third- and fourth-instar stages led to misshapen pupal setae. The setae were shortened or became black nodules. Treatment with ds*dyl* at the larval and pupal stages resulted in deformed adults with completely suppressed wing hairs. Moreover, the knockdown of *Hvdyl* at the third-instar stage caused deformed larval mouthparts at the fourth-instar period. As a result, foliage consumption was inhibited, and larval growth was slowed. The results indicate that Dyl is associated with the growth of cellular protuberances throughout development and with the formation of the cuticle in *H. vigintioctopunctata*.

## 1. Introduction

Dusky-like (Dyl) is a member of the zona pellucida (ZP) domain family, a group of proteins related to linking the insect cuticle to the plasma membrane of epidermal cells, regulating the cytoskeleton and deposition of cuticle [1,2,3,4,5,6]. In *Drosophila melanogaster*, Dyl has been documented to be a transmembrane and secreted protein [3,7]. It is involved in the organization of the apical extracellular domain in the epidermis and the deposition of chitin. As a result, a deficiency in Dyl impairs the formation of embryonic denticles, adult sensory bristles, and wing hairs [3,5,8]. 

Dyl also plays a crucial role in the development of the Coleoptera *Tribolium castaneum* [3,6,9]. Dyl in *T. castaneum* reaches peak expression in the late embryonic stage, and RNAi of *Tcdyl* in the parents leads to significant embryonic lethality. Depletion of Dyl results in defects in larval epidermal pigmentation and completely blocks the transitions from larval-to-pupal and pupal-to-adult stages [9].

It appears that Dyl has undergone a functional shift between Diptera and Coleoptera insects. Further investigation of Dyl in other insects will be helpful to further clarify its function and help us understand its role in insect growth and development. *Henosepilachna vigintioctopunctata* (Coleoptera: Coccinellidae), a destructive pest that attacks potato, tomato, eggplant, and Chinese boxthorn, possesses several kinds of cellular protuberances. The larvae have two types of extensions. Firstly, an *H. vigintioctopunctata* larva has 11 rows of dark-branched spines (scoli) located on brown epidermal mastoids (strumae) [10], similar to its sibling species [11]. Secondly, sensory setae are located on the larval mouthparts and legs, the pupal thorax, and the adult mouthparts and elytra. Furthermore, *H. vigintioctopunctata* is sensitive to RNA interference (RNAi) [10,12,13,14,15]. This offers an ideal insect model to investigate the roles of Dyl throughout the entire lifecycle. 

In this study, we characterized *Hvdyl* in *H. vigintioctopunctata* using RNAi. We observed two characteristic RNAi phenotypes seen in *D. melanogaster* and *T. castaneum*. Firstly, the knockdown of *Hvdyl* inhibited the growth of cellular protuberances, a defect observed in the *D. melanogaster* [3,5,8]. Secondly, the knockdown of *Hvdyl* inhibited foliage consumption and affected the survival of the beetles, a major phenotype described in Dyl RNAi experiments in *T. castaneum* [9]. We argued that the link between the cuticle and the plasma membrane of epidermal cells is disturbed in the Dyl-deficient *H. vigintioctopunctata* larvae. This represses the development of cellular protuberances and blocks the formation of the cuticle. As a result, food ingestion is reduced, and survival is affected.

## 2. Methods and Materials

### 2.1. Insect

*H. vigintioctopunctata* was routinely maintained in an insectary at 25 ± 1 °C under a 16 h:8 h light–dark photoperiod and 50–60% relative humidity with potato foliage at vegetative growth or tuber initiation stages provided as food.

### 2.2. Molecular Cloning

The *Hvdyl* sequence was obtained from *H. vigintioctopunctata* transcriptome data. The total RNA was extracted using TRIzol reagent (Invitrogen, New York, NY, USA) in accordance with the manufacturer’s protocols. The RNA was quantified with a NanoDrop 2000 spectrophotometer (Thermo Fisher Scientific, New York, NY, USA). RNA purity was determined by assessing optical density (OD) absorbance ratios at OD260/280 and OD260/230. Polymerase chain reaction (PCR) was used to verify the sequences. The primers are listed in Appendix A. The cDNA sequence was submitted to GenBank (accession number: *Hvdyl*, OP056326).

### 2.3. Preparation of dsRNAs

We cloned the cDNA fragments of *Hvdyl* and an enhanced green fluorescent protein (*egfp*). *Hvdyl* and *egfp* were amplified with PCR using specific primers (Appendix A) conjugated with the T7 RNA polymerase promoter. Two independent ds*Hvdyl* primers were designed (ds*Hvdyl*-F1R1, ds*Hvdyl*-F2R2) (Appendix A, Appendix A). Due to the consistent RNAi phenotypes observed with both sets of primers, the results based on primary primers (ds*Hvdyl*-F1R1) are presented in this article (Appendix A). In order to verify whether there are any possible off-target sequences that possess an identical match of 20 bp or more, we used BLAST to search these two targeted regions against the *H. vigintioctopunctata* transcriptome. The dsRNAs (ds*dyl*, ds*egfp*) were synthesized using the MEGAscript T7 High Yield Transcription Kit (Ambion, Austin, TX, USA) in accordance with a previously described protocol [16]. The quality of the dsRNA was evaluated via agarose gel electrophoresis, and the concentration was quantified with a Nanodrop 1000 spectrophotometer.

### 2.4. Injection of dsRNA

A liquid drop (0.1 μL), including 300 ng ds*dyl* or ds*egfp,* was injected into newly ecdysed third- or fourth-instar larvae or 1-day-old pupae [17,18]. Each replicate included 10 beetles, and each treatment was repeated 6 times. Three of these repeats were used to collect samples (2 days after the injection) and measure RNAi efficacy, while the other three groups were used to observe phenotypes over the next three weeks. 

### 2.5. Observation of Feeding Ability

Potato leaves of similar sizes were collected and cut into a circle with a radius of 5 mm and placed in Petri dishes. A beetle that had been injected with ds*egfp* or ds*dyl* was transferred to the dish. Three larvae were set as a replicate, and each treatment was repeated 6 times. The gnawed leaves were measured after 24 h. The larval mouthparts were observed, and the larvae were dissected to image their guts.

### 2.6. Quantitative Real-Time PCR (qRT-PCR)

Beacon Designer 7 (Premier Biosoft International, Palo Alto, CA, USA) was used to design the qRT-PCR primers given in Appendix A. The RNA was extracted from the collected samples and reverse-transcribed into cDNA. *HvRPS18* and *HvRPL13* were selected as control genes according to previous reports (the primers are listed in Appendix A) [19]. The qRT-PCR protocol consisted of 1 cycle of 95 °C for 30 s, followed by 40 cycles of 95 °C for 5 s, and then 60 °C for 34 s. In order to verify the PCR products, the heat dissociation curves and amplification plots were analyzed at the end of the thermal cycle. The data were analyzed via the 2^−ΔΔCT^ method. Each experiment was repeated 3 times.

## 3. Results

### 3.1. Dyl Is Conserved in Insects

We cloned the full-length Dyl cDNA and analyzed its sequence alignment and evolutionary relationships. Sequence alignment of the proteins encoded with Dyl from *H. vigintioctopunctata*, *D. melanogaster*, and *T. castaneum* demonstrated that Dyl codes for a transmembrane protein containing a ZP domain (Figure 1A). 

Phylogenetic analysis revealed that Dyl had an orthologue in each insect and showed a one-to-one orthologous relationship among insects. Dyl proteins from Coleoptera, Lepidopteran, Hymenoptera, and Diptera species formed four clusters. As expected, *Hvdyl* belongs to the Coleoptera group (Figure 1B).

### 3.2. Expression Profiles of Hvdyl

The mRNA levels of *Hvdyl* during different developmental stages were measured using qRT-PCR. *Hvdyl* transcript levels could be detected in embryos, larvae, prepupae, pupae, and adults. It was highly expressed during the embryonic period, at late stages of the first-, second-, and third-instar stages, and at the pupal period. Conversely, *Hvdyl* had low transcription levels in fourth-instar larvae and newly eclosed adults (Figure 2A).

The tissue-based expression of *Hvdyl* was also investigated. *Hvdyl* was broadly expressed in the epidermis, foregut, midgut, hindgut, Malpighian tubules, and body fat of 1-day-old third-instar larvae. The highest expression level was found in the epidermis, followed by the midgut, Malpighian tubules, foregut, and hindgut; the lowest level was detected in the body fat (Figure 2B). 

### 3.3. Hvdyl RNAi at the Third-Instar Stage Inhibited the Growth of Scoli and Setae

*Hvdyl* was knocked down with RNAi in the third-(penultimate)-instar larvae (Figure 3A). The *Hvdyl* RNAi larvae could molt to the fourth-instar larvae (Figure 3B). However, two obvious phenotypic defects were observed (Figure 3C–H). Firstly, 11 rows of branched spines (scoli) are located on the epidermal mastoids (struma) of an *H. vigintioctopunctata* larva. In dsegfp-injected beetles, the scoli were strong and highly branched, and a spine was present on the tips (Figure 3C,E). In contrast, the larval scoli were undeveloped, and the tip spines were missing in the *Hvdyl* RNAi larvae (Figure 3D,F). Moreover, the scoli were weak and soft, and some were bending (Figure 3D). Secondly, the setae on the larval head capsule and mouthparts did not develop, with only small dots remaining in the *Hvdyl* RNAi larvae (Figure 3H,G). 

Approximately 80% of the fourth-instar *Hvdyl* RNAi larvae were pupated successfully (Figure 3I). Knockdown of *Hvdyl* negatively affected the growth of pupal setae. The pupal setae formed black tubercles. Moreover, the bald scoli were attached to the abdomen in the *Hvdyl* RNAi pupae (Figure 3K,J). 

Most of the *Hvdyl* knockdown pupae emerged to form deformed adults (Figure 3L). About 50% of the *Hvdyl*-depleted adults had deformed wings, and around 20% were wrapped in the old pupal cuticles (Figure 3L,N,O). These misshapen adults hardly moved and fed and eventually died within 7 days after molting (Figure 3N,O). 

### 3.4. Reduced Growth and Foliage Consumption in the Hvdyl RNAi Beetles

The *Hvdyl*-depleted beetles were weighed 0, 2, 3, 6, 9, and 14 days after injection of ds*dyl* at the third-instar larval stage. The fresh weights of the *Hvdyl* RNAi beetles were statistically significantly lower than those of the ds*egfp*-injected controls 3, 6, 9, and 14 days after ds*dyl* injection (Figure 4A). Consistently, the 1- and 3-day-old *Hvdyl*-silenced fourth-instar larvae, pupae, and adults were smaller than their ds*egfp*-injected counterparts (Figure 4B–E). 

The foliage consumption of the fourth-instar *Hvdyl* RNAi larvae was examined. One day after ecdysis, the larvae in the ds*egfp*-treated group had eaten all the leaves, while the larvae in the ds*dyl*-injected group only ate about 30% of the leaves (Figure 4F). The guts of almost all ds*egfp*-treated larvae were full of food (Figure 4G). In contrast, there was almost no food in the guts of the ds*dyl*-treated larvae (Figure 4H). 

At 30 min (Figure 4I,J) and 2 h (Figure 4K,L), after the ds*dyl*-injected larvae molted to fourth instars, the larval mouthparts were examined. In ds*egfp*-treated larvae, the mandibles on both sides met and were well pigmented (Figure 4I,K). Conversely, the mandibles of the ds*dyl*-injected larvae were weakly tanned and separated from each other (Figure 4J,L).

### 3.5. Knockdown of Hvdyl at the Fourth-Instar Stage

Two days post-injection of ds*dyl* into newly molted fourth-instar larvae, the target gene was markedly down-regulated (Figure 5A). Similarly to the control larvae, almost all ds*dyl*-treated larvae were pupated successfully (Figure 5B). However, the growth of the setae on the thoraces of pupae was completely inhibited, and black nodules were formed (Figure 5D). About 10% of the *Hvdyl* RNAi pupae emerged to form normal adults. The remaining 90%, however, exhibited phenotypic defects (Figure 5C,E). The elytra and hindwings of these ds*dyl*-treated adults were much smaller in size than those of control adults. The elytra were improperly folded and wrapped with pupal cuticles; the membranous wings were completely shrunken, and the tips of the membranous wings became withered and darkened (Figure 5F). In addition, the hairs on the elytra were suppressed in the ds*dyl*-treated adults (Figure 5F).

### 3.6. Depletion of Hvdyl at the Pupal Stage

Similarly, injection of ds*dyl* into newly molted pupae successfully knocked down *Hvdyl* (Figure 6A). All ds*dyl*-treated pupae formed defective adults (Figure 6B). Compared with the control adults, the ds*dyl*-treated adults were completely wrapped with pupal exuviae (Figure 6C,D). The elytra in the pupal exuviae were improperly folded. After the removal of the pupal cuticle, it was obvious that the development of hairs on the elytra was also inhibited. The membranous wings were shrunken, and the tips became withered and darkened (Figure 6E,F).

## 4. Discussion

In this study, we discovered that two characteristic Dyl RNAi phenotypes seen in *D. melanogaster* [3,5,8] and *T. castaneum* [9] were also observed in *Henosepilachna vigintioctopunctata*. We discuss these phenotypes below.

### 4.1. Dyl Is Critical for the Development of Epidermic Cellular Protuberances

In this study, we provided four pieces of experimental evidence to support the requirement for Dyl in the growth of cellular protuberances.

Firstly, the tissue-specific analysis revealed that the highest expression level of *Hvdyl* was in the epidermis, where the scoli and setae developed in *H. vigintioctopunctata* (Figure 2B). Consistently, Dyl in *D. melanogaster* is detected in the trichome cells of the embryonic epidermis [3,6]. During post-embryonic development, Dyl protein accumulates in wing hairs and the body hairs [5]. Tissue expression patterns suggest that Dyl functions in the cuticle, especially in the cellular protuberances.

Secondly, RNAi of *Hvdyl* in third-instar larvae caused undeveloped larval scoli throughout the whole body and truncated setae on the larval head capsule and mouthparts in *H. vigintioctopunctata* (Figure 3). Similarly, different ZP proteins localize to different parts of the first-instar larval denticles and play a specific role in the morphogenesis of denticles [3]. In *D. melanogaster* embryos, a mutation in Dyl causes very small, unhooked, severely split denticles [3]. Consistently, the knockdown of Dyl leads to very small denticles in embryos [5]. These findings provide conclusive experimental evidence to confirm the role of Dyl in the control of the development of insect cellular protuberances.

Thirdly, the knockdown of *Hvdyl* led to abnormal setae on the pupal thoraces in *H. vigintioctopunctata* (Figure 3K and Figure 5D). In *T. castaneum*, RNAi of *Tcdyl* at the larval stage completely blocks the larval–pupal transition, and at the pupal period, it inhibits the pupal–adult transformation [9]. Therefore, we, for the first time, have observed a pupal Dyl RNAi phenotype: living pupae with impaired thorax setae.

Fourthly, the depletion of *Hvdyl* brought about misshapen bristles on the adult wings of *H. vigintioctopunctata* (Figure 5E,F). In line with our results, Dyl function is necessary to form adult sensory bristles and wing hairs in *D. melanogaster* [5,8]. Knocking down Dyl in adults results in stubby bristles associated with bristle collapse [5]. Moreover, the depletion of Dyl in adults causes split, thinned, and often very short wing hairs [8].

All these pieces of experimental evidence confirm that Dyl is required for the growth of cellular protuberances in Coleoptera insects, similar to its role in *D. melanogaster* [5,8].

Actin filaments have been proven to be critical for the shape and elongation of the cellular protuberances [5,20,21,22,23,24,25]. Once the elongation of cellular protuberances has finished, the shape is believed to be maintained by the chitinous exoskeleton [26]. Therefore, a shortage of actin or chitin can cause misshapen epidermal cellular protuberances. Does the impairment of cellular protuberances in *Hvdyl* RNAi *H. vigintioctopunctata* beetles result from actin shortage, chitin deficiency, or both?

### 4.2. Dyl May Regulate Chitin Deposition

Here, we first discuss a lack of chitin in the *Hvdyl* RNAi beetles in *H. vigintioctopunctata*. It is known that *Drosophila* Dyl and other ZP domain proteins are associated with chitin deposition [3,5]. In our study, we provided five pieces of experimental evidence to demonstrate that Dyl is associated with the formation of the cuticle, possibly controlling chitin deposition. 

Firstly, we noted that *Hvdyl* was highly expressed during the embryonic period, at late stages of first-, second-, and third-instar larvae and in pupae (Figure 2A). Our data suggest that Dyl is necessary for the formation of the cuticle after each ecdysis. This is consistent with observations in *D. melanogaster,* where the expression of Dyl increases 118-fold from the early to late stages in the process of hair morphogenesis [27]. Interestingly, the mRNA levels of *UDP-N-acetylglucosamine pyrophosphorylase* (*UAP*) and *chitin synthase 1* (*CHS1*) involved in chitin biosynthesis are periodically raised during ecdysis and reduced in the active feeding stage in *H. vigintioctopunctata* [28,29] and *L. decemlineata* [30,31]. It appears that Dyl may be involved in the regulation of chitin production after each ecdysis, similar to *D. melanogaster* [8].

Secondly, the knockdown of *Hvdyl* caused deformed and weakly tanned larval mouthparts in *H. vigintioctopunctata* (Figure 4J,L). Similar phenotypic defects have been documented when chitin synthesis has been repressed in *H. vigintioctopunctata*, where the clypeus, labrum, mandibles, maxillae, and labium in the mouthparts were weakly tanned, and the mandibles could not meet each other, in contrast to controls [28]. In *H. vigintioctopunctata* [28] and *L. decemlineata* [31], the shortage of chitin also slightly changes the adult pigmentation in other organs, especially the coloration on the elytra. In *D. melanogaster*, loss of Dyl brings about abnormal pigmentation of the bristles [5,8]. Moreover, the epidermis in the *Tcdyl* RNAi animals in *T. castaneum* is poorly pigmented [9]. According to the results in other Coleoptera [28,29,31] and Diptera [5,8] species, the poorly pigmented epidermis in the *Tcdyl* RNAi animals in *T. castaneum* [9] may also be due to a lack of chitin.

Thirdly, the knockdown of *Hvdyl* reduced foliage consumption in *H. vigintioctopunctata* (Figure 4F). As a result, growth was inhibited (Figure 4B–E). In accordance with our results, disruption of the chitin biosynthesis pathway has been shown to negatively affect feeding in *H. vigintioctopunctata* [28,29]. Normal settling and feeding on host plant leaves depend on normal larval mouthparts [32]. Dysfunction of larval mouthparts in the *Hvdyl* (this study), *HvCHS1,* or *HvUAP* [28,29] RNAi larvae undoubtedly affects food ingestion. 

Fourthly, the scoli were weak and soft, and some were bending in the *Hvdyl* RNAi larvae (Figure 3D). Similar phenotypic defects of scoli are observed when chitin biosynthesis is repressed by RNAi of *UAP* in *H. vigintioctopunctata* [29]. 

Lastly, injection of ds*dyl* at the larval and pupal stages resulted in deformed adults (Figure 3N,O, Figure 4E, Figure 5E and Figure 6D). In accordance with our results, the depletion of the *HvCHS1* [28] and *HvUAP* [29] bring about misshapen beetles in *H. vigintioctopunctata*. In *T. castaneum*, all *Tcdyl* RNAi larvae fail to pupate. Moreover, injection of ds*Tcdyl* into early-stage pupae results in developmental arrest prior to eclosion in the *T. castaneum* [9]. The phenotypic differences between *T. castaneum* and *H. vigintioctopunctata* may be explained by different RNAi efficacies. In fact, knockdown of *HvUAP* can arrest the larval development at the third- or fourth-instar larval or prepupal stages [29].

It appears that in both Coleoptera [9] and Diptera [8] species, Dyl is involved in chitin accumulation in the cuticle. Depletion of Dyl inhibits the deposition of chitin and, subsequently, the coloration of the cuticle [8,9]. In *D. melanogaster*, Dyl functions as a Rab11 effector for the chitin deposition [8]. Whether the same pathway is present in *H. vigintioctopunctata* needs further investigation to clarify.

### 4.3. Chitin Is Not the Only Target of Dyl

A lack of chitin in *H. vigintioctopunctata* only caused soft scoli but did not affect the morphogenesis of the setae [29]. Obviously, a deficiency in chitin does not mimic the Dyl-defective phenotypes of the epidermal cellular protuberances. Similarly, *Drosophila* Dyl mutant phenotypes in wing hairs are distinctly different from and are more severe than the phenotypes caused by a deficiency in the chitin [5,8]. Therefore, a defect in chitin deposition cannot explain all of the *Hvdyl* RNAi defective phenotypes. Dyl must have other targets in the morphogenesis of the cellular protuberances in *H. vigintioctopunctata*. 

In terms of this context, the chitin-binding protein Knickkopf (Knk) is believed to play an important role in protecting the new cuticle from degradation by chitinases [33,34]. Additionally, a kind of epidermal endochitinase from *T. castaneum,* TcCht7, is also considered to play a role in organizing chitin in the newly forming cuticle rather than in degrading the chitin present in the old cuticle [35]. Moreover, Chitinase 6 (Cht6) is a strong candidate for a Dyl target in *D. melanogaster* because depletion of *Cht6* brings about all of the Dyl hair phenotypes, albeit to a weaker extent [8]. Whether a similar signal pathway is involved in *H. vigintioctopunctata* remains to be clarified.

Given that a deficiency of chitin cannot fully explain the Dyl RNAi phenotype, the morphological abnormalities of epidermal cellular protuberances in the *Hvdyl* knockdown may be due to the effects on the cytoskeleton. In this context, *Drosophila* Dyl and other ZP domain proteins have been suggested to be able to regulate the cytoskeleton [1,2,6,8]. Specifically, the knockdown of Dyl in wing cells causes premature hair initiation [8]. Accordingly, Dyl was found to negatively control actin deposition before hair initiation in hair-forming cells [8]. However, the regulative network of actin is very complicated in *Drosophila*. For instance, mutations in the key general component genes *cofilin* and *AIP1* [36,37], in F-actin bundling protein genes *forked* and *singed* [20,23,38], in non-muscle myosin encoding genes *crinkled* [39] and *zipper* [40], in regulator gene *Rho kinase* [41], and in F-actin-depolymerizing protein-encoding genes *twin star* [36] and *flare* [37] result in hair phenotypes. Whether the actin-related cytoskeleton components are affected in the *Hvdyl* RNAi beetles is another interesting question that deserves investigation in the future.

## 5. Conclusions

In summary, deficiency in Dyl disturbs the link between the cuticle and the plasma membrane of epidermal cells in *H. vigintioctopunctata* larvae. This represses the development of cellular protuberances and blocks the formation of the cuticle. As a result, food ingestion is reduced, and survival is affected.

## Figures and Tables

**Figure 1 biology-12-00866-f001:**
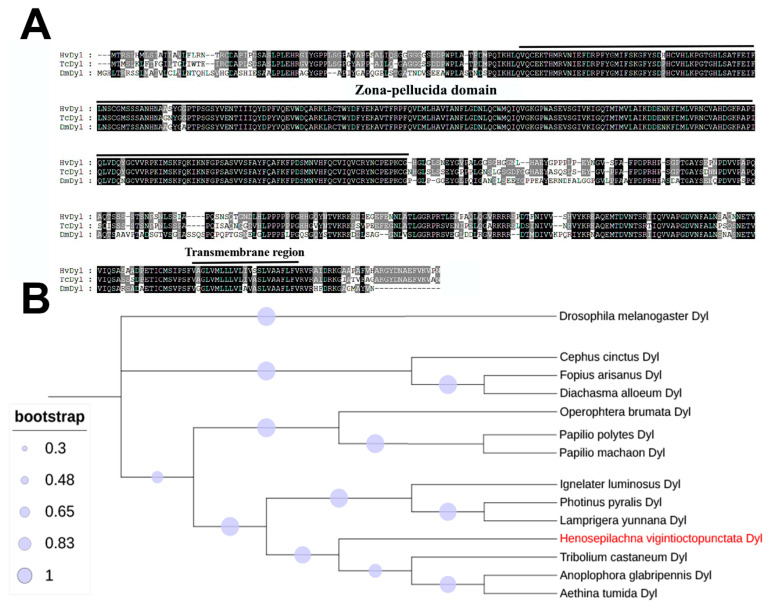
Alignment and phylogenetic analysis of dusky-like (Dyl). (**A**) Sequence alignment of Dyl proteins. The ZP domain and transmembrane region are marked. Increasing background intensity (from light to dark) indicates an increase in sequence similarity. Gaps are introduced to permit alignment. (**B**) The tree was constructed using the neighbor-joining method. Bootstrap analyses of 1000 replications were carried out. The Dyl proteins are from seven Coleoptera [*H. vigintioctopunctata* (*Hvdyl*) (Red label), *Aethina tumida* (AtDyl, XM_020015731.1), *Anoplophora glabripennis* (*Ag*Dyl, XM_018722972.1), *T. castaneum* (*Tc*Dyl, XM_008201614.2), *Ignelater luminosus* (*Il*Dyl, KAF2884998.1), *Lamprigera yunnanaand* (*Ly*Dyl, KAF5283461.1), and *Photinus pyralis* (*Pp*Dyl, XP_031337078.1)], three Lepidopteran [*Operophtera brumata* (*Ob*Dyl, KOB66544.1), *Papilio machaon* (*Pm*Dyl, XM_014511564.1), and *Papilio polytes* (*Pp*Dyl, XM_013279810.1)], three Hymenopteran [*Cephus cinctus* (*Cc*Dyl, XM_015745504.2), *Diachasma alloeum* (*Da*Dyl, XM_015266059.1), and *Fopius arisanus* (*Fa*Dyl, XM_011302635.1)], and one Diptera [*D. melanogaster* (*Dm*Dyl, AAF47884.1)].

**Figure 2 biology-12-00866-f002:**
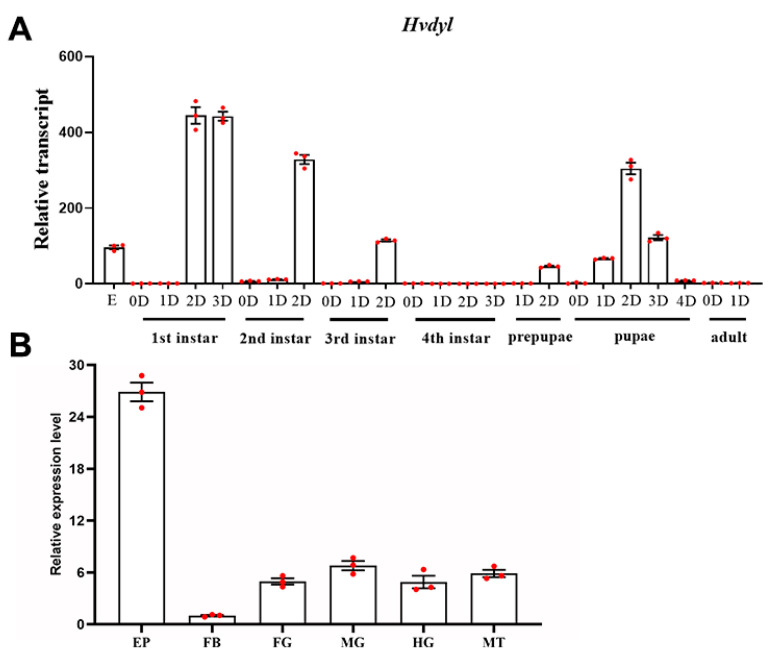
The transcription patterns of *Hvdyl* in *Henosepilachna vigintioctopunctata*. For temporal expression analysis (**A**), RNA templates were derived from eggs (day 3); the larvae from the first through to the fourth instars, prepupae, pupae, and adults (0D indicates newly ecdysed larvae or pupae or newly emerged adults). For tissue expression analysis (**B**), the relative transcripts were measured in the foregut (FG), midgut (MG), hindgut (HG), Malpighian tubules (MT), epidermis (dorsal) (EP), and body fat (FB) of the day 2 third-instar larvae. For each sample, three independent pools of 20–30 individuals were measured in technical triplicates using quantitative real-time polymerase chain reaction. The three biological replicates are marked with red dots. The values were calculated using the 2^−ΔΔCT^ method. The lowest transcript level of mRNA at a specific developmental time point or tissue was set as 1. The columns represent averages with vertical lines indicating SE.

**Figure 3 biology-12-00866-f003:**
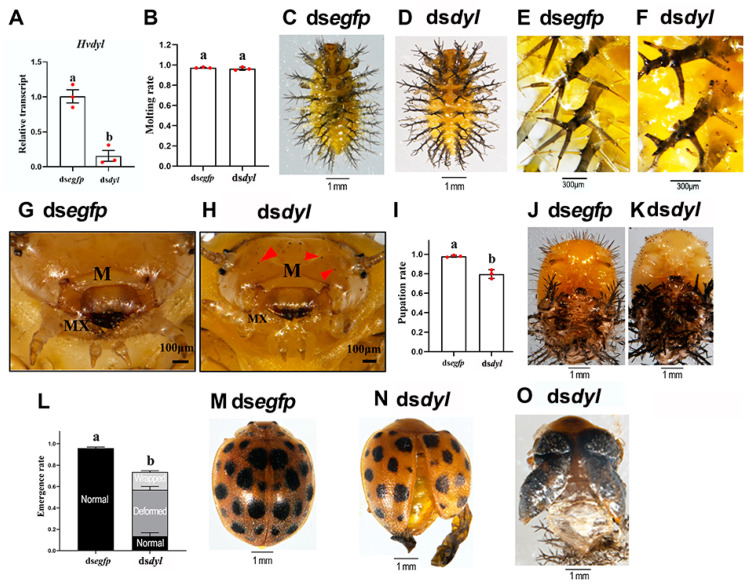
RNAi of *Hvdyl* impairs growth of the cellular protuberances and the formation of cuticle in *Henosepilachna vigintioctopunctata*. The newly molted third-instar larvae were injected with 300 ng ds*egfp* or ds*dyl*. The larvae were then transferred to potato foliage. Two days after injection, transcript levels of *Hvdyl* (**A**) were measured. Relative transcript levels are the ratios of relative copy number in treated individuals to that of ds*egfp*-injected controls, which was set as 1. The three biological replicates are marked with red dots. The larval molt, pupation, and emergence rates were recorded during a 2-week trial period (**B**,**I**,**L**). The values represent means (±SE). Different letters indicate significant differences at *p* < 0.05 using *t*-test. The phenotype of fourth-instar larvae (**C**–**H**) (The red triangle indicates defective bristles), pupae (**J**,**K**), and adults (**M**–**O**) are shown. M, mouthparts; MX, maxillae.

**Figure 4 biology-12-00866-f004:**
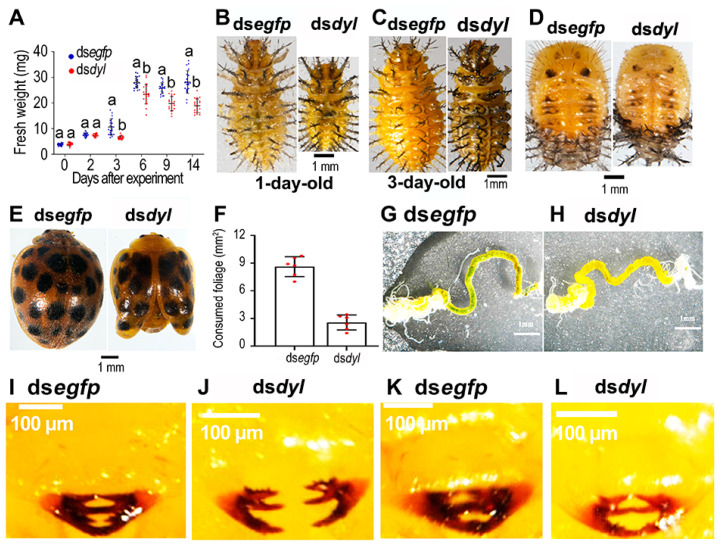
The reduced foliage consumption and body sizes of the *Hvdyl* RNAi *Henosepilachna vigintioctopunctata* beetles. The ds*egfp*- and ds*dyl*-injected larvae were weighed during a 2-week trial period (**A**). The foliage consumption at 24 h post molting was measured (**F**). The data are represented as values (±SE). Different letters indicate significant differences at *p* < 0.05 using *t*-test. The whole 1- and 3-day-old fourth-instar larvae, pupae, and adults were imaged (**B**–**E**). The guts of the fourth-instar larvae that ingested leaves for 24 h were dissected and imaged (**G**,**H**). Dorsal views of the mouthparts of the fourth-instar larvae at 30 min (**I**,**J**) and 2 h post molting are shown (**K**,**L**).

**Figure 5 biology-12-00866-f005:**
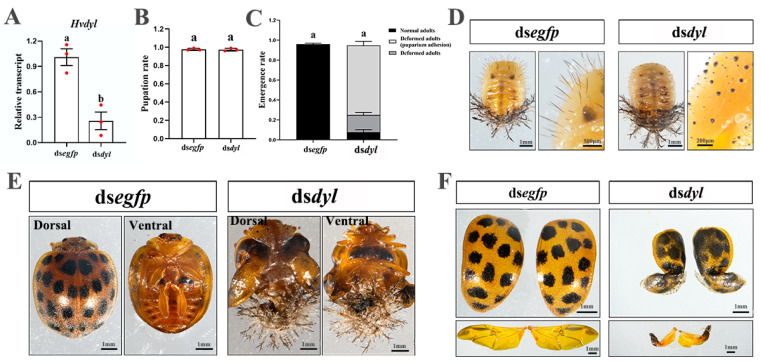
Knockdown of *Hvdyl* at the fourth-instar larval stage causes abnormal *Henosepilachna vigintioctopunctata* pupae and adults. The newly molted fourth-instar larvae were injected with 300 ng ds*egfp* or ds*dyl* and then transferred to potato foliage. Two days after injection, transcript levels of *Hvdyl* were measured (**A**). Relative transcript levels are the ratios of relative copy number in treated individuals to that of ds*egfp*-injected controls, which was set as 1. The pupation and emergence rates were recorded during a 2-week trial period (**B**,**C**). The values represent means (±SE). Different letters indicate significant differences at *p* < 0.05 using *t*-test. The three biological replicates are marked with red dots. The phenotypes of pupae (**D**), adults (**E**), elytra, and hindwings (**F**) were observed and imaged.

**Figure 6 biology-12-00866-f006:**
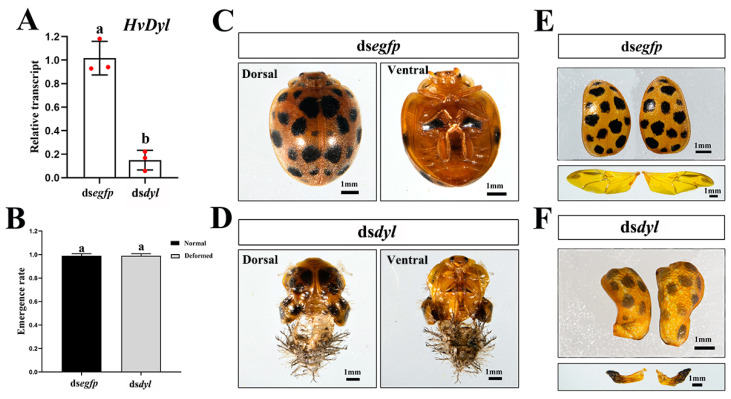
Depletion of *Hvdyl* at pupae stage results in misshapen *Henosepilachna vigintioctopunctata* adults. The newly molted pupae were injected with 300 ng ds*egfp* or ds*dyl*. Two days after injection, the transcript levels of *Hvdyl* were measured (**A**). Relative transcript levels are the ratios of relative copy number in treated individuals to that of ds*egfp*-injected controls, which was set as 1. The emergence rates were recorded during a 1-week trial period (**B**). The values represent means (±SE). Different letters indicate significant differences at *p* < 0.05 using *t*-test. The three biological replicates are marked with red dots. The phenotypes of adults (**C**,**D**), elytra, and hindwings (**E**,**F**) were observed and imaged.

## Data Availability

Data generated in association with this study are available in the Appendix A published online with this article.

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
