# Peer review of "Dusky-like Is Critical for Morphogenesis of the Cellular Protuberances and Formation of the Cuticle in Henosepilachna vigintioctopunctata"

_biology, 2023, doi:10.3390/biology12060866_

Round 1
Reviewer 1 Report
Dusky-like (Dyl) is a member of the zona pellucida domain family of transmembrane protein, which has been reported to function on formation of cuticular protuberances. Authors studies Dyl function in a phytophagous beetle, Henosepilachna vigintioctopunctata. They found that Dyl was expressed in epidermis before each ecdysis. Knockdown of Dyl by dsRNA injection into larvae and pupae showed defects on formation of cuticular protuberances including dorsal scoli of larvae and bristles and on wing formation. These data indicate that Dyl is involved in formation of cellular protuberances and cuticle in this species. The current study provides a set of valuable data to understand the function of Dyl in insects. This reviewer feels several concerns as follows:
1. Conclusion of Dyl function
Authors describes that “our findings indicated that Dyl has occurred functional shift between Dipteran and Coleopteran 20 insects” in the simple summary, but this point is not discussed in the main text. In the conclusion section, authors pointed out Dyl function for linking of the cuticle and the plasma membrane. Although Dyl is a membrane protein, authors do not provide any evidence to support it.
2. RNAi experiment
Authors likely used only a single construct of dsRNA. Even though authors checked possible off-target of this construct, it is not proved. It is required to use another dsRNA designed to target another region of Dyl mRNA and examine whether it shows same phenotype. It is also necessary to show the target region on Dyl mRNA of dsRNA.
Other minor concerns are listed below:
1. Title
The expression of “intracellular extensions” is obscure. “Cellular protuberances” or “cellular extensions” seem clearer. “Epidermal protuberances” might be good, too. Same expression is found at other sections.
2. line 31
The explanation for the abbreviation, dsDyl, is required for the first time.
3. Figure 1
On Fig. 1A, the region of “Zona-pellucida super family” is indicated. This should be “Zona-pellucida domain”. Fig. 1A is an alignment of three sequences. But the legend for Fig. 1A describes 12 sequences.
4. Expression analysis (Figure 2)
To reveal the distribution of Dyl mRNA, authors used several tissues of day-1 third-instar larvae (Fig. 2B), which expressed Dyl at lower level (Fig. 2A). This reviewer wonders why authors did not used tissues of stages with higher Dyl expression.
5. Figure 2B
The region of epidermis used in this analysis should be stated.
6. Figure 3
The scale bars of Fig. 3EF are indicated 1 mm. These might be incorrect. In Fig. 3G, it is difficult to see the setae on dsegfp-injected larvae. “M” and “MX” should be explained in the legend.
7. lines 224-225, Figure 4I-L
Authors describes that the mandibles met each other in the control animals but not in the RNAi animals. Because mandibles often open and close, the photo for the mandibles in Fig. 4J was possibly taken just when they opened. In Fig. 4L, the mandibles of the RNAi larvae looked closed. It is described that Fig. 4IJ were 30 min and Fig. 4KL were two hours after molting in the main text, but this information should be stated in the figure legend, too.
8. Figure 5AB
The control does not indicate “1”, it looks around “0.95”.
9. Figure 5E and 6CD
It should be stated the figures are dorsal views or ventral views.
10. line 248
Although it is described that dsDyl was injected into day-1 pupae, the figure legend says that newly molted pupae were injected.
11. Discussion
Some discussions are confusing. Especially, the logic or point of discussion behind the line 284-288 cannot be seen to this reviewer.
As described at the previous section, the logics and points of explanation of some paragraphs are difficult to understand. Several spelling and grammatical mistakes were also found. Please fix them.
Reviewer 2 Report
The authors present a rather simply designed but thoroughly performed and well-illustrated study on the role of Dusky-like (Dyl) gene in the development and morphology of a polyphagous beetle, Henosepilachna vigintioctopunctata. I would say it is of interest to entomologists. The classic RNAi and qRT-PCR methods were used to investigate Dyl expression in different tissues and developmental stages and the effect of Dyl knockdown on morphology of fourth instar larvae, pupae and adults of H. vigintioctopunctata.
The main problem that I see with this manuscript is bad English. I am not a native speaker, but I can tell that there are many spelling and grammatical errors in the text. Some of these errors сould be detected by any standard grammar software. At the same time, I believe that the manuscript needs professional English editing in terms of the grammatical structure of the sentences.
Major comments.
1) I would say that the last sentence in “Simple Summary” in unjustified.
2) “Methods and Materials” section is unnecessarily concise:
- PCR conditions are not described;
- it is not described exactly how the injection was carried out;
- the methods of bioinformatics and statistical analysis applied are not described at all.
Minor comments.
Chapter 3.4. is not formatted as a chapter title.
Comments on the Quality of English Language.
1) In line 31, I believe 'eperdermic' should be ‘epidermal’.
2) In line 162, I think ‘frome’ is a typo and it should be ‘from’.
3) In lines 201-202, I believe 'statistically significant lower' should be ‘significantly lower’.
4) In lines 223, 224, 323, ‘mandbles’ should be ‘mandibles’.
5) In line 331, ‘neccessory’ should be ‘necessary’.
Comments on the Quality of English Language.
1) In line 31, I believe 'eperdermic' should be ‘epidermal’.
2) In line 162, I think ‘frome’ is a typo and it should be ‘from’.
3) In lines 201-202, I believe 'statistically significant lower' should be ‘significantly lower’.
4) In lines 223, 224, 323, ‘mandbles’ should be ‘mandibles’.
5) In line 331, ‘neccessory’ should be ‘necessary’.
Reviewer 3 Report
Authors show that RNAi knockdown of the Dusky-like gene in a polyphagous beetle inhibited the growth of extracellular extensions (and thus, for example the growth of mandibles), which resulted in an inhibition of feeding, and finally in survival of the beetles. The role of of Dyl in beetles is not novel, and a functional shift of Dyl between Diptera and Coleoptera had also been shown before. Nevertheless, the present study is professionally done and can be published in Biology, Section Developmental Biology. The manuscripts, however, needs considerable editorial and language improvement.
- either use Diptera and Coleoptera in uppercase letters, or coleopterans and dipterans in lowercase letters
- use ml or mL etc. in the entire manuscript
- line 22: zona pellucida is the correct name
- line 40: H. vigintioctopunctata is the correct name
- line 72: RNAi of
- line 80: and potato....as food
- headline 3.4 in italics
- line 223 and others: mandibles is the correct spelling
- 4.2: that Dyl regulates chitin deposition in the present case is speculation and should be said so (see lines 358/359)
- References must be presented in a uniform style and according to MDPI Authors' Instructions
Minor language corrections are necessary, especially the use of singular and plural.
Reviewer 4 Report
In this manuscript, the authors conduct an RNAi experiment to knockdown expression of dusky-like and identify phenotypes compared to a control (gfp). The authors compare the phenotypes to those published for D. melanogaster and T. castaneum.
The authors do a good job setting up the RNAi experiments and assays. Clearly defects are present in the Dyl knockdowns. The figures are nice.
My main critics are as follows:
1. the authors have missed a big section of the literature on cuticle formation etc. in T. castaneum. Look for papers from the Beeman and Arakane groups.
2. That said, the discussion on T. castaneum could also be improved.
3. The authors say it remains unknown if the pathways between H. vigintioctopunctata and D. melanogaster are the same and that it would be interesting to see if Dyl also acts on the cytoskeleton of H. vigintioctopunctata. I suggest an RNAseq experiment comparing Dyl RNAi individuals compared to those in injected. It is a relatively easy way to see what other genes are affected and what pathways might be at play when Dyl is knocked down.
There are several places where the incorrect tense of the verb is used, wrong word used, or words are misspelled. Examples:
Line 56. It should read "It appears that there as been a functional shift in Dyl between Dipteran and Coleopteran insects" or something similar.
Line 66 should read "... throughout all developmental stages" or just " throughout development.
Line 75- " As a result, digestion of food is reduced and survival rate is lower"
There are many more examples. The author may want to have a native English speaker proofread the manuscript.